# Structural insights into SETD3-mediated histidine methylation on β-actin

Qiong Guo[1†], Shanhui Liao[1†*], Sebastian Kwiatkowski[2†], Weronika Tomaka[2], Huijuan Yu[1], Gao Wu[1], Xiaoming Tu[1], Jinrong Min[3,4], Jakub Drozak[2*], Chao Xu[1*]

[1]Division of Molecular and Cellular Biophysics, Hefei National Laboratory for Physical Sciences at the Microscale, School of Life Sciences, University of Science and Technology of China, Hefei, China; [2]Department of Metabolic Regulation, Faculty of Biology, University of Warsaw, Warsaw, Poland; [3]Structural Genomics Consortium, University of Toronto, Toronto, Canada; [4]Department of Physiology, University of Toronto, Toronto, Canada

**Abstract** SETD3 is a member of the SET (Su(var)3–9, Enhancer of zeste, and Trithorax) domain protein superfamily and plays important roles in hypoxic pulmonary hypertension, muscle differentiation, and carcinogenesis. Previously, we identified SETD3 as the actin-specific methyltransferase that methylates the N3 of His73 on β-actin (Kwiatkowski et al., 2018). Here, we present two structures of S-adenosyl-L-homocysteine-bound SETD3 in complex with either an unmodified β-actin peptide or its His-methylated variant. Structural analyses, supported by biochemical experiments and enzyme activity assays, indicate that the recognition and methylation of β-actin by SETD3 are highly sequence specific, and that both SETD3 and β-actin adopt pronounced conformational changes upon binding to each other. In conclusion, this study is the first to show a catalytic mechanism of SETD3-mediated histidine methylation on β-actin, which not only throws light on the protein histidine methylation phenomenon but also facilitates the design of small molecule inhibitors of SETD3.
DOI: https://doi.org/10.7554/eLife.43676.001

**\*For correspondence:**
ajsod@mail.ustc.edu.cn (SL);
jdrozak@biol.uw.edu.pl (JD);
xuchaor@ustc.edu.cn (CX)

[†]These authors contributed equally to this work

**Competing interests:** The authors declare that no competing interests exist.

## Introduction

Microfilaments are the building blocks of the cytoskeleton and are made up of actin proteins (*dos Remedios et al., 2003*; *Theriot and Mitchison, 1991*). There are six actin isoforms in mammalian cells that are characterized on the basis of their different expression profiles and cellular functions, including $\alpha_{skeletal}$-, $\alpha_{cardiac}$-, $\alpha_{smooth}$-, $\beta_{cyto}$-, $\gamma_{cyto}$-, and $\gamma_{smooth}$-actins (*Gunning et al., 1983*; *Herman, 1993*; *Perrin and Ervasti, 2010*). Among these, β-actin is ubiquitously expressed and plays critical roles in a wide variety of cellular functions, such as cytoskeleton formation, cell motility and maintenance of cell stability (*Leterrier et al., 2017*; *Nudel et al., 1983*).

Many different types of post-translational modifications (PTMs) have been found in actin proteins, including acetylation, methylation, SUMOylation and ubiquitination (*Terman and Kashina, 2013*). $N^3$-methylation of His73 in β-actin has been of special interest since its identification in 1967 (*Johnson et al., 1967*), not only because of the extreme evolutionary conservation of this PTM– methylated His, which is present in almost all eukaryotic actins, but mainly because the methylation of H73 decreases the hydrolysis rate of the actin-bound ATP (*Kabsch et al., 1990*), as evidenced by the greater ATP exchange rate in the actin-H73A mutant (*Nyman et al., 2002*).

Although the presence of actin-specific methyltransferases was shown in rabbit muscles decades ago (*Raghavan et al., 1992*; *Vijayasarathy and Rao, 1987*), the molecular identity of this enzyme has long been unknown. Recently, we and others identified SETD3 as the actin-specific histidine N-methyltransferase that methylates actin at His73 (*Kwiatkowski et al., 2018*; *Wilkinson et al., 2019*).

SETD3 is a member of the SET domain family, the SET domain being a ~ 130 amino acids (aa) motif initially identified in *Drosophila* proteins, with its acronym derived from three histone lysine methyltransferases: **S**u(var)3–9, **E**nhancer of zeste, and **T**rithorax (*Dillon et al., 2005*). To date, most known histone lysine methyltransferases contain a SET domain, although there are some exceptions such as DOT1L (*Min et al., 2003*), METTL12 (*Rhein et al., 2017*), and METTL21B (*Malecki et al., 2017*). Before SETD3 was identified as an actin-specific histidine methyltransferase, SET domain proteins were known to act as lysine methyltransferases, transferring a methyl moiety from the methyl donor S-adenosyl methionine (AdoMet) to the substrates to generate a methylated form of the substrate, with *S*-adenosyl-L-homocysteine (AdoHcy) as the co-product (*Dillon et al., 2005*).

As (i) SETD3 has been identified previously as a histone methyltransferase that methylates histone H3 at Lys4 and Lys36 (*Eom et al., 2011*; *Wagner and Carpenter, 2012*) and as (ii) the results of our preliminary experiments using isothermal calorimetry titration (ITC) and mass spectrometry had revealed that SETD3 binds to and methylates a β-actin peptide containing His73 (66–88) (but not H3K4 (1–23) or H3K36 (25–47) peptides), we crystallized and solved two structures of AdoHcy-bound SETD3 in complex with either the His73 peptide or the 3-methylhistidine (His73me) peptide to determine the molecular mechanism of SETD3-mediated histidine methylation of β-actin.

Having solved the two peptide-bound structures of SETD3, we reveal that SETD3 recognizes a fragment of β-actin in a sequence-dependent manner and utilizes a specific pocket to catalyze the $N^3$-methylation of His73. Moreover, our comprehensive structural, biochemical and enzymatic profiling of SETD3 allows us to pinpoint the key residues in SETD3 that are important for substrate recognition and subsequent methylation. Therefore, our structural research, supplemented by biochemical and enzymatic experiments, not only provides insights into the catalytic mechanism of SETD3 but will also facilitate the design of specific inhibitors of the SETD3 enzyme.

## Results

### SETD3 binds to and methylates β-actin

We and others had previously shown that SETD3 acts as an actin-specific histidine *N*-methyltransferase, but the molecular basis for the selective histidine methylation catalyzed by SETD3 remained unknown (*Kwiatkowski et al., 2018*; *Wilkinson et al., 2019*). We measured the binding affinity of the SETD3 core region (aa 2–502) to a His73-containing fragment of β-actin (aa 66–88) by ITC and found that SETD3 binds to the β-actin peptide with a Kd of 0.17 μM (*Figure 1A–B* and *Table 1*). Given that SETD3 has also been reported to be a putative lysine methyltransferase that methylates Lys4 and Lys36 of histone H3 (*Eom et al., 2011*), we also examined the binding of SETD3 to two different histone peptides, H3K4(1–23) and H3K36(25–47), and found that neither of these peptides binds to SETD3 (*Table 1*).

Furthermore, we detected the activity of SETD3 on β-actin(66–88), H3K4(1–23), and H3K36(25–47) by mass spectrometry and found that SETD3 methylates the β-actin peptide (*Figure 1—figure supplement 1A*) but does not modify either H3K4 or H3K36 (*Figure 1—figure supplement 1B–1C*). No methylated product was detected for any of the three peptides in the presence of AdoMet without the addition of SETD3 (*Figure 1—figure supplement 1A–1C*). Moreover, SETD3 did not methylate the β-actin(66–88) H73A mutant, although binding of the mutant peptide to SETD3 is only ~2.5 fold weaker than that of the wild-type peptide (0.45 μM vs. 0.17 μM) (*Figure 1B* and *Figure 1—figure supplement 1D*). Collectively, the binding experiments and mass spectrometry data indicate that SETD3 specifically binds to the β-actin peptide and methylates it at His73.

### Overall structure of SETD3

To uncover the mechanisms underlying the recognition and methylation of β-actin by SETD3, we attempted to crystallize AdoMet-bound SETD3 with full-length β-actin, but failed to obtain diffractable crystals. By using the core region of SETD3(2–502) with the β-actin peptide(66–88), we succeeded in crystallizing the complex and obtained a 1.95 Å structure of AdoHcy-bound SETD3(2–502) with β-actin(66–88) (*Table 2*, *Figure 1—figure supplement 2A*, *Figure 1—figure supplement 3*). There are four molecules in a crystallographic asymmetric unit and residues 22–501 of SETD3 and

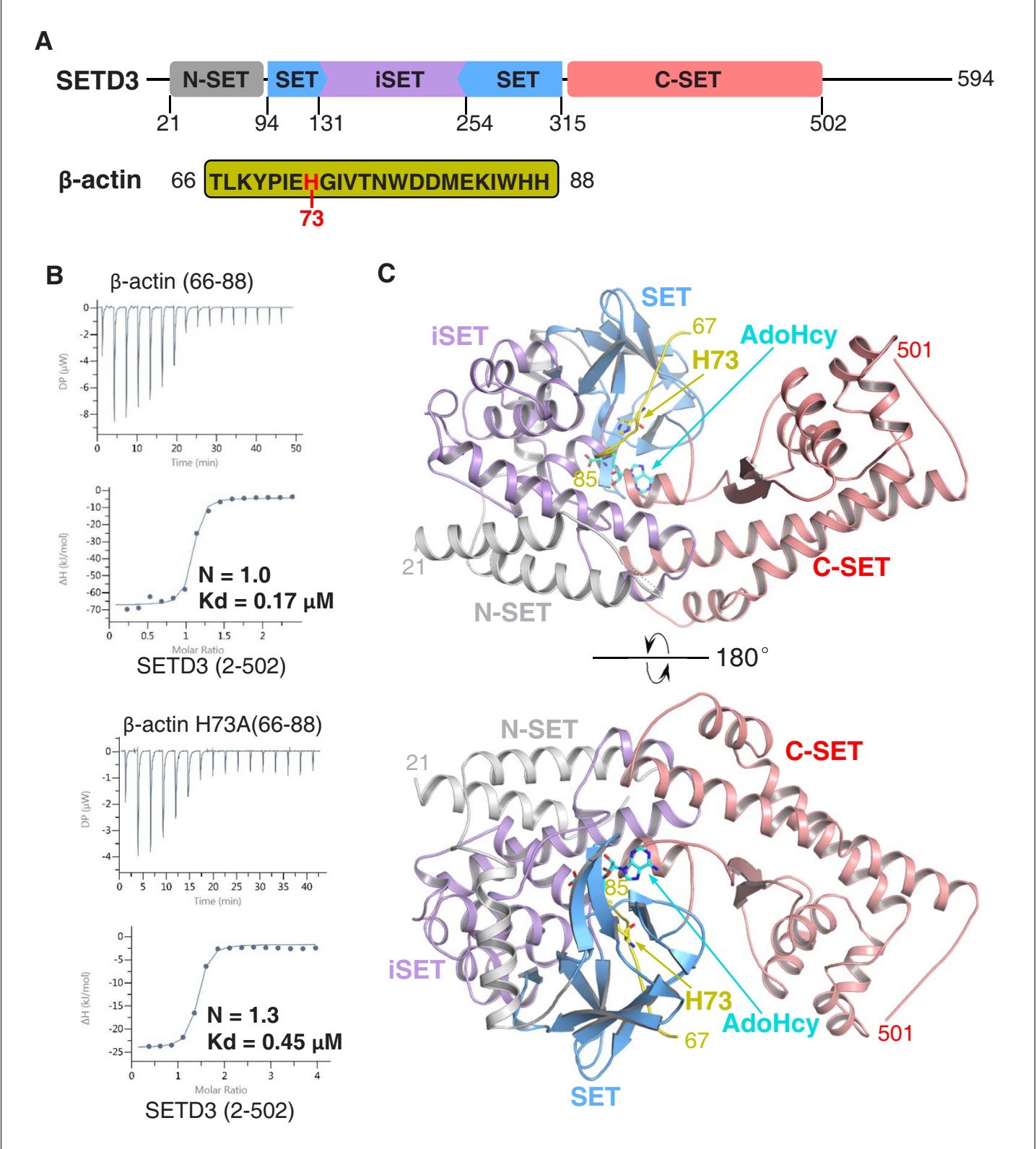

**Figure 1.** The SETD3 core region specifically recognizes a fragment of β-actin containing His73. (**A**) Domain architecture of full-length human SETD3 (aa 1–594) and the sequence of the β-actin peptide (66–88), with His73 of β-actin highlighted. (**B**) Representative ITC binding curves for the binding of SETD3 (aa 2–502) to β-actin peptides of different lengths. The molecular ratios, derived Kds and respective standard deviations are also indicated. (**C**) The overall structure of AdoHcy-bound SETD3 with unmodified β-actin peptide. The SETD3 domains are colored in the same way as in *Figure 1A*, with

*Figure 1 continued on next page*

*Figure 1 continued*

the N-SET, SET, iSET and C-SET regions of SETD3 colored in gray, blue, purple and pink, respectively. The peptide is shown in yellow cartoon, while His73 and AdoHcy are shown in yellow and cyan sticks, respectively. His73 of actin and the AdoHcy are labeled.

DOI: https://doi.org/10.7554/eLife.43676.002

The following figure supplements are available for figure 1:

**Figure supplement 1.** Mass Spectrometry data showing the activity of SETD3 toward different peptides.

DOI: https://doi.org/10.7554/eLife.43676.003

**Figure supplement 2.** Sequence alignment of human SETD3 with its orthologs or other SET domain proteins (SETD6 and LSMT).

DOI: https://doi.org/10.7554/eLife.43676.004

**Figure supplement 3.** Topology of SETD3 (aa 2–502), with secondary structures marked and colored as in *Figure 1—figure supplement 2A*.

DOI: https://doi.org/10.7554/eLife.43676.005

**Figure supplement 4.** Superposition of SETD3 molecules.

DOI: https://doi.org/10.7554/eLife.43676.006

**Figure supplement 5.** The 2|Fo|–|Fc| σ-weighted maps of peptide and AdoHcy.

DOI: https://doi.org/10.7554/eLife.43676.007

**Figure supplement 6.** Comparison of AdoHcy's mode of binding to SETD3, LSMT or SETD6.

DOI: https://doi.org/10.7554/eLife.43676.008

**Figure supplement 7.** Superposition of AdoMet-bound SETD3 (PDB id: 3SMT) and AdoHcy-bound SETD3-actin.

DOI: https://doi.org/10.7554/eLife.43676.009

**Figure supplement 8.** The overall structure of SETD3 with methylated β-actin peptide in the presence of AdoHcy.

DOI: https://doi.org/10.7554/eLife.43676.010

66–84 of β-actin are visible in the complex structures (*Figure 1—figure supplement 4A*). On the basis of the density map, we found that His73 of β-actin had been methylated in the structure (*Figure 1—figure supplement 5A*), although we used an unmethylated peptide. The reason for this could be that SETD3 bound to the methyl donor AdoMet from *Escherichia coli*, which we used to

**Table 1.** Binding affinities of SETD3 proteins to β-actin peptides.

| β-actin peptide | SETD3 (aa 2–502) | Stoichiometric coefficient (N) | Kd (μM) |
|---|---|---|---|
| [66]TLKYPIEHGIVTNWDDMEKIWHH[88] | Wildtype | 1.0 | 0.17 ± 0.04 |
| [66]TLKYPIEAGIVTNWDDMEKIWHH[88] (H73A) | Wildtype | 1.3 | 0.45 ± 0.08 |
| [1]ARTKQTARKSTGGKAPRKQLATK[23] (H3K4) | Wildtype | *N/A | *NB |
| [25]ARKSAPATGGVKKPHRYRPGTVA[47] (H3K36) | Wildtype | N/A | NB |
| [66]TLKYPIEHGIVTNWDDMEKIWHH[88] | R215A | 1.2 | 3.6 ± 1.0 |
| [66]TLKYPIEHGIVTNWDDMEKIWHH[88] | N256A | 0.94 | 2.1 ± 0.4 |
| [66]TLKYPIEHGIVTNWDDMEKIWHH[88] | N256D | 1.3 | 23 ± 7 |
| [66]TLKYPIEHGIVTNWDDMEKIWHH[88] | N256Q | 1.1 | 2.5 ± 0.5 |
| [66]TLKYPIEHGIVTNWDDMEKIWHH[88] | W274A | 0.94 | 0.51 ± 0.10 |
| [66]TLKYPIEHGIVTNWDDMEKIWHH[88] | Y313F | 0.86 | 3.0 ± 0.1 |
| [66]TLKYPIEHGIVTNWDDMEKIWHH[88] | R316A | 1.1 | 7.4 ± 0.9 |
| [66]TLKAPIEHGIVTNWDDMEKIWHH[88] (Y69A) | Wildtype | 1.4 | 3.8 ± 0.8 |
| [66]TLKYPAEHGIVTNWDDMEKIWHH[88] (I71A) | Wildtype | 1.3 | 16 ± 3 |
| [66]TLKYPIEHGIVTNWADMEKIWHH[88] (D80A) | Wildtype | 1.3 | 3.5 ± 0.4 |
| [66]TLKYPIEHGIVTNWDAMEKIWHH[88] (D81A) | Wildtype | 1.3 | 2.0 ± 0.5 |
| [66]TLKYPIEHGIVTNWDDAEKIWHH[88] (M82A) | Wildtype | 1.2 | 0.89 ± 0.08 |
| [66]TLKYPIEHGIVTNWD[80] | Wildtype | 1.0 | 2.9 ± 0.7 |

*N/A: Not applicable. *NB: no binding affinity detectable by ITC.

Dissociation constants (Kds) were derived from a minimum of two experiments (mean ± S.D.). The original binding curves are shown in *Supplementary file 1*.

DOI: https://doi.org/10.7554/eLife.43676.011

**Table 2.** Data collection and refinement statistics.

| | SETD3–β-actin | SETD3-methylated β-actin |
|---|---|---|
| Protein Data Bank ID | 6ICV | 6ICT |
| Data collection | | |
| Radiation wavelength (Å) | 0.9789 | 0.9792 |
| Space group | P 1 21 1 | P 1 21 1 |
| Cell dimensions | | |
| $a$, $b$, $c$ (Å) | 60.19, 175.17, 66.50 | 59.98, 176.69, 125.89 |
| $\alpha$, $\beta$, $\gamma$ (°) | 90, 92.57, 90 | 90, 93.37, 90 |
| Resolution (Å) | 35.00–2.15 (2.23–2.15) | 72.27–1.95 (2.06–1.95) |
| $R_{merge}$ | 0.135 (0.609) | 0.169 (0.868) |
| $I / \sigma I$ | 14 (3.4) | 8.1 (2.6) |
| CC1/2 | 0.988 (0.884) | 0.991 (0.827) |
| Completeness (%) | 100(100) | 99.1 (99.7) |
| Redundancy | 6.4 (6.4) | 7.0 (7.0) |
| Refinement | | |
| Resolution (Å) | 35.00–2.15 | 59.87–1.95 |
| No. of reflections (used/free) | 71998/3629 | 187163/9084 |
| $R_{work}$/$R_{free}$ | 0.168/0.205 | 0.178/0.214 |
| Number of atoms/B-factor (Å$^2$) | 8493/30.7 | 16956/33.7 |
| Protein | 7574/30.3 | 15147/33.1 |
| Peptide | 276/32.9 | 591/38.4 |
| AdoHcy | 52/19.3 | 104/20.0 |
| Solvent | 591/36.2 | 1114/40.0 |
| RMSD bonds (Å)/angles (°) | 0.007/0.84 | 0.007/0.87 |
| Ramachandran Plot favored/allowed/outliers (%) | 98.76/1.24/0 | 98.98/1.02/0 |

Values in parentheses are for the highest-resolution shell.
DOI: https://doi.org/10.7554/eLife.43676.012

express our protein, and that the AdoMet-bound SETD3 methylated the peptide at His73 during crystallization. Therefore, this complex represents a snapshot of the post- methyl transfer state, which prompted us to crystallize the substrate-bound enzyme.

To avoid the methylation reaction, SETD3 was purified by including a 5-fold excess of AdoHcy in the buffer that would compete with AdoMet for binding with SETD3. Then, the non-methylated peptide was mixed with the purified SETD3 to form the complex. The complex was crystallized and solved at a resolution of 2.15 Å (*Table 2*). There are two molecules in a crystallographic asymmetric unit (*Figure 1—figure supplement 4B*). In this structure, the density map of the peptide indicates that it is unmodified, suggesting that methylation did not occur (*Figure 1—figure supplement 5B*). The complex structure therefore reflects the image of the pre-methyl transfer state.

In both substrate-bound and product-bound structures, the SETD3 core region adopts a V-shape architecture and is composed of N-SET (N-terminal to the SET domain), SET, iSET (an insertion in the SET domain), and C-SET (C-terminal to the SET domain) domains (*Figure 1C*). The N-terminal lobe contains the N-SET (α1–α3), SET (β1–β12), and iSET (α4–α11) domains (*Figure 1—figure supplement 2A*, *Figure 1—figure supplement 3*). The N-SET domain consists of three α helices (α1–α 3), and encompasses the SET and iSET domains (*Figure 1—figure supplement 2A*, *Figure 1—figure supplement 3*). α1 and α2 of N-SET form a four-helix bundle with α10–α11 of iSET, whereas α3 of N-SET is localized at the SET-iSET interface and comes into contact with the anti-parallel β strands (β1–β2) of SET and α8 of iSET (*Figure 1C*, *Figure 1—figure supplement 3*). The SET domain adopts a canonical SET domain fold similar to that of LSMT (*Trievel et al., 2002*), with twelve β strands arranged into four anti-parallel β sheets (β1-β2, β3-β11, β4-β10-β9, β5-β7-β6) and one parallel β

sheet (β8-β12) (*Figure 1—figure supplement 2A*, *Figure 1—figure supplement 3*). The iSET domain is a helical region (α4–α11) inserted between β5 and β6 of the SET domain (*Figure 1—figure supplement 2A*, *Figure 1—figure supplement 3*). The C-terminal lobe, consisting of just the C-SET domain, is almost entirely helical (α12–α19) except for two anti-parallel strands (β13–β14), with α12 and the N-terminal end of α19 packed against α10 and α9 of iSET, respectively (*Figure 1—figure supplement 2A*, *Figure 1—figure supplement 3*). Consistently, a database search using the DALI server (*Holm and Rosenström, 2010*) revealed that the overall fold of SETD3 is highly similar to those of LSMT (Z-core: 31.4, RMSD: 3.8 Å for 425 Cα atoms) and SETD6 (Z-core: 28.7, RMSD: 2.8 Å for 421 Cα atoms), despite the low sequence identities (24–25%).

## AdoHcy-binding pocket within SETD3

In both substrate-bound and product-bound structures, AdoHcy is located at a cleft in SETD3 formed by the SET and C-SETD domains, and is buttressed by iSET at the bottom (*Figure 1C*). Specifically, the adenine ring of AdoHcy is sandwiched between the side chain of Glu104 and the aromatic ring of Phe327, forming stacking and π–π interactions with Glu104 and Phe327, respectively (*Figure 1—figure supplement 6A*). The $N^6$ and $N^7$ atoms of AdoHcy are hydrogen-bonded to the main chain carbonyl and amide groups of His279, respectively, while the AdoHcy $C^8$ atom forms one C-H...O hydrogen bond with the hydroxyl group of Tyr313 (*Figure 1—figure supplement 6A*).

In addition to the adenine-specific interactions, the ribose O3′ and O4′ atoms of AdoHcy form hydrogen bonds with the main chain carbonyl of Ser325 and the side chain carboxyl of Asn278, respectively (*Figure 1—figure supplement 6A*). The side chain carboxyl of Asn278 makes another hydrogen bond with the AdoHcy amide, and the latter is also H-bonded to the main chain carbonyl of Phe106 (*Figure 1—figure supplement 6A*). The AdoHcy carboxylate group forms four hydrogen bonds with SETD3 residues: one with the main chain amide of Phe106, one with the guanidino group of Arg254, and the other two with the guanidino group of Arg75 (*Figure 1—figure supplement 6A*). All AdoHcy-binding residues are conserved in other SETD3 orthologs, suggesting a conserved function of SETD3 (*Figure 1—figure supplement 2A*).

The AdoHcy-binding mode of SETD3 is similar to those observed in some other SET domain structures, such as those of LSMT (*Trievel et al., 2002*) and SETD6 (*Chang et al., 2011*) (*Figure 1—figure supplement 6B–6C*). Among these AdoHcy-binding residues, Asn278, His279 and Phe327 are conserved in LSMT and SETD6, whereas Glu104 and Arg254 are conserved only in LSMT and not in SETD6 (*Figure 1—figure supplement 2B*, *Figure 1—figure supplement 6B–6C*). Arg75 is an AdoHcy-binding residue that is found only in SETD3 (*Figure 1—figure supplement 2B*).

## β-actin-binding mode of SETD3

In both substrate-bound and product-bound structures, the β-actin peptide lies in a narrow groove formed by SET, iSET and C-SET, with the unmodified or methylated His73 residue snugly accommodated in a pocket composed of α11 of iSET, β6 and β12 of SET, and α12 of C-SET, and η5 of SET (*Figure 1C*, *Figure 2A*). When superimposing our substrate-bound complex structure with the previously determined AdoMet–SETD3 binary structure (PDB id: 3SMT), we found that, despite the overall structural similarity between the two SETD3 structures (with a RMSD of 0.66 Å in Cα positions), the region containing two β-sheets of the SET domain (β4-β10-β9 and β3-β11) adopts a conformational change upon peptide binding. Specifically, peptide binding induces the three loops preceding β4, β9 and β11 to shift toward the peptide by 4.2 Å, 4.4 Å and 6.8 Å, respectively (*Figure 1—figure supplement 7*). The detailed interactions between SETD3 and the N-terminal end of the peptide will be described in the following section.

## A unique histidine-recognition pocket in SETD3 confers its histidine methyltransferase activity

In the substrate-bound structure, residues Leu67-Glu83 of the β-actin peptide are visible, with His73 occupying a hydrophobic pocket (*Figure 2A*). The main chain of His73 forms several hydrogen bonds with SETD3, with its main chain amide, Cα, and main chain carbonyl H-bonded to the main chain carbonyl of Tyr313, the side chain carboxyl of Asn256, and the guanidino of Arg316, respectively (*Figure 2A*). The imidazole ring of His73 is parallel to the aromatic ring of Tyr313, with its orientation determined by two hydrogen bonds: one between the $N^1$ atom of the ring and the

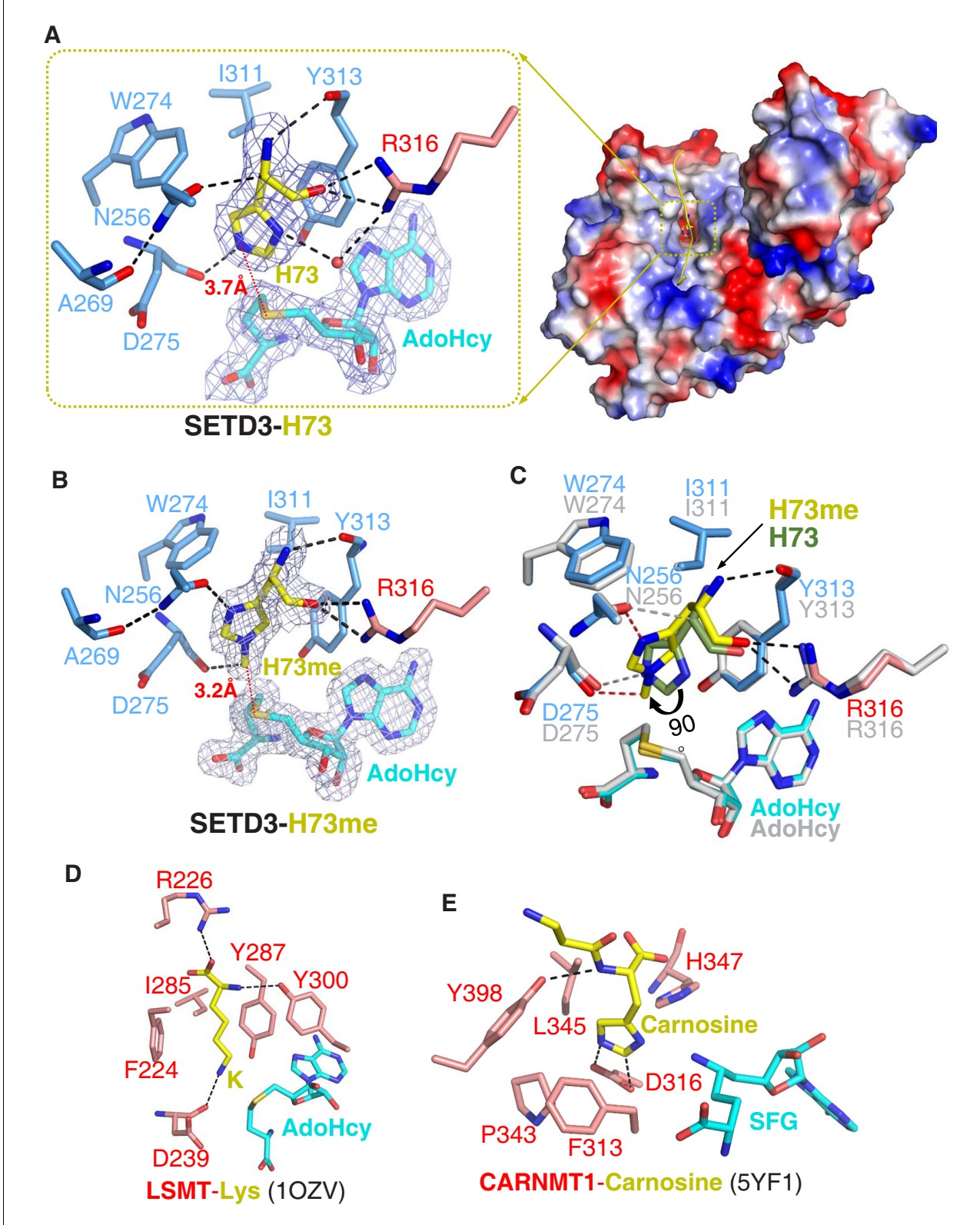

**Figure 2.** Molecular mechanism of substrate recognition and histidine methylation by SETD3. (**A**) The β-actin peptide binds into a long groove on the surface of the SETD3 N-lobe region (right), with His73 of β-actin positioned into a hydrophobic pocket (left). His73 and AdoHcy are shown in yellow and cyan sticks, respectively, and their 2|Fo|–|Fc| σ-weighted maps are contoured at 1.2 σ. The His73-binding residues of SETD3 and are colored according to on regions in which they reside, as shown in *Figure 1A*. (**B**) Detailed interactions between His73me and SETD3 in the post- methyl transfer
*Figure 2 continued on next page*

*Figure 2 continued*

complex. His73me and AdoHcy are shown in yellow and cyan sticks, respectively, and their 2|Fo|–|Fc| σ-weighted maps are contoured at 1.2 σ. The His73me-binding residues of SETD3 are colored according to the scheme shown in *Figure 2A*. (C) Superposition of the two complexes on the histidine/methylhistidine binding pocket. For the SETD3–His73me complex, the His73me-binding residues of SETD3 are illustrated as in *Figure 2A*, with His73me and AdoHcy shown in yellow and cyan sticks, respectively. For the SETD3–His73 complex, the His73-binding residues and AdoHcy are shown in gray sticks, whereas His73 is shown in green sticks to show that it rotates 90 degrees after catalysis. After methylation, one new hydrogen bond is formed between Asn256 and the $N^1$ atom of His73me (red dashed line). (D) Lysine recognition by LSMT(PDB id: 1OZV) in the presence of AdoHcy. (E) Carnosine recognition by CARNMT1 (PDB id: 5YF1) in the presence of the AdoHcy mimics SINEFUNGIN (SFG). Histidine, lysine and carnosine are shown in yellow sticks, whereas the protein residues that are involved in binding are shown in red sticks. AdoHcy and SFG are shown in cyan sticks.

DOI: https://doi.org/10.7554/eLife.43676.013

The following figure supplements are available for figure 2:

**Figure supplement 1.** Detailed interactions of SETD3 and β-actin peptide(66–88).
DOI: https://doi.org/10.7554/eLife.43676.014

**Figure supplement 2.** Interactions between SETD3 and peptide.
DOI: https://doi.org/10.7554/eLife.43676.015

**Figure supplement 3.** Actin undergoes conformational changes upon binding to SETD3.
DOI: https://doi.org/10.7554/eLife.43676.016

guanidino of Arg316 mediated by a water molecule, and the other between the $N^3$ atom of the ring and the main chain carbonyl of Asp275 (*Figure 2A*). In addition to the stacking interactions with Tyr313, His73 also makes hydrophobic contacts with Trp274 and Ile311. The distance between the $N^3$ atom of His73 and the sulfur atom of AdoHcy is 3.7 Å, suggesting that this structure represents the pre-methyl transfer state well (*Figure 2A*).

In the product complex, the residues Leu67-Lys84 and the main chain of Ile85 of the β-actin peptide are visible, with His73 methylated to His73me (*Figure 1—figure supplement 8*). Despite the fact that the methylated peptide adopts a $3^{10}$ helix at its C-terminal end, the substrate-bound and product complexes are highly similar, with an RMSD of 0.19 Å over protein Cα atoms and an RMSD of 0.32 Å over peptide Cα atoms, respectively. In the product-bound structure, His73me is inserted into the same pocket of SETD3 as in the substrate complex but the distance between the installed methyl group of His73 and the sulfur atom of AdoHcy is 3.2 Å, with its density map distinguishable from that of unmodified histidine (*Figure 2B*).

Superposition of the substrate-bound and product complexes clearly indicates that the hydrogen bonds between the main chain of His73me and the main chain carbonyl of Tyr313 and the guanidino of Arg316 are intact after methylation (*Figure 2B*). However, the imidazole ring of His73me is rotated by ~90° relative to that of the unmodified His73, with its $C^4H$ pointing toward the ring of Tyr313 (*Figure 2B–2C*) and its $N^1$ atom hydrogen-bonded to the side chain carboxyl group of Asn256 (*Figure 2B*).

When comparing SETD3 with the other SET-domain-containing lysine methyltransferases, we found that there are two main reasons that SETD3 is not likely to be a lysine methyltransferase. First, the histidine-binding pocket of SETD3 is too shallow to accommodate the aliphatic side chain of a lysine. Replacement of His73 with a lysine would not only disrupt the water-mediated hydrogen bond between $N^1$ of His73 and Arg316, but could also lead to a steric clash between the aliphatic side chain of lysine and the main chain of the bottom carbonyl cage residue, Asp275. Second, the phenylalanine of LSMT and SETD6, which creates a cation-π interaction with the lysine, is replaced by an asparagine (Asn256) in SETD3, and Asn256 is critical for substrate binding and histidine methylation (*Figure 2A–2B* and *Figure 1—figure supplement 2A*). Therefore, this substitution establishes an imidazole ring-specific interaction with histidine by impairing the lysine-specific interaction.

## SETD3 exhibits extensive interactions with β-actin

In both substrate-bound and product complexes, peptide residues flanking His73 or His73me of β-actin also interact extensively with the N-terminal lobe of SETD3, with Leu67-Glu72 and Gly74-Glu83 mainly contacting the SET and iSET motifs, respectively (*Figure 2—figure supplement 1*, *Figure 2—figure supplement 2*). Leu67 and Pro70 of β-actin make hydrophobic contacts with Ile284 of SETD3 and there are additional hydrophobic interactions between Tyr69 of β-actin and Pro259, Tyr288 and

Leu290 of SETD3 (*Figure 2—figure supplement 1*, *Figure 2—figure supplement 2*). Two main chain hydrogen bonds are formed between Tyr69 of β-actin and Tyr288 of SETD3, and between Ile71 of β-actin and Thr286 of SETD3, inducing the shift of the two sheets (β4-β10-β9 and β3-β11) in SETD3, as mentioned above (*Figure 1—figure supplement 7*). Ile71 of actin is positioned into a hydrophobic pocket of SETD3 that is formed by Ile258, Ile271, Trp274, Thr286, Tyr288, and Cys295 (*Figure 2—figure supplement 1*, *Figure 2—figure supplement 2*). Notably, Leu283-Thr285 of SETD3 adopts a β strand in the structure of AdoMet-bound SETD3 without a peptide (PDB: 3SMT), but decomposes upon binding to the N-terminal side of β-actin, which shortens the β9 strand of SETD3 in the peptide-bound structure (*Figure 1—figure supplement 7*). The side chain carboxylate group of Glu72 forms one hydrogen bond with the guanidino group of Arg316, which is the only C-SET domain residue that contacts the N-terminal side of the β-actin peptide (*Figure 2—figure supplement 1*, *Figure 2—figure supplement 2*).

Regarding the residues of the C-terminal to His73 of β-actin, the main chain amide group of Gly74 is hydrogen-bonded to the main chain carbonyl group of Gln255, while the side chain of Val76 stacks with the imidazole ring of His324 (*Figure 2—figure supplement 1*, *Figure 2—figure supplement 2*). In contrast to Gly74 and Val76, the Ile75 side chain is solvent-exposed and is therefore not involved in the interactions with SETD3 (*Figure 2—figure supplement 1*). Thr77 not only makes hydrophobic contact with Leu268, but also forms two hydrogen bonds through its hydroxyl group with the side chain amide groups of Asn154 and Gln255 (*Figure 2—figure supplement 1*, *Figure 2—figure supplement 2*). The side chain amide group of Asn154 makes another hydrogen bond with the main chain carbonyl of Asn78. The indole ring of Trp79 is inserted into an open hydrophobic pocket composed of Ile155, Val248, Val251 and Met252, with its main chain carbonyl H-bonded to the side chain amide of Gln216, while the carboxyl group of Asp80 is hydrogen-bonded to the side chain amide of Arg215 (*Figure 2—figure supplement 1*, *Figure 2—figure supplement 2*). Asp81 of β-actin forms three hydrogen bonds with SETD3: one with the side chain amide of Gln216 and two with the guanidino group of Arg215 (*Figure 2—figure supplement 1*, *Figure 2—figure supplement 2*). Notably, the β-actin peptide sequence is also conserved in α-and γ-actin proteins, suggesting that SETD3 might also recognize and methylate other actin isotypes.

## Key SETD3 residues in β-actin binding and methylation

To evaluate the roles of the SETD3 residues around the active site in β-actin methylation, we made several single SETD3 mutants and quantitatively compared their affinities for binding to the β-actin peptide by ITC. All of the SETD3 mutants displayed weaker binding affinities toward the peptide, but R215A and R316A exhibited the most significantly reduced binding affinities: 21-and 42-fold, respectively (*Table 1*). Our complex structures showed that both Arg215 and Arg316 make several hydrogen bonds with β-actin (*Figure 2B*, *Figure 2—figure supplement 1*), implying that they play a critical role in substrate recognition (*Table 1*). N256A binds the peptide with a 12-fold lower affinity than wildtype SETD3, (Kd = 2.1 µM) (*Table 1*); this is because the mutation disrupts not only the hydrogen bond to the main chain of the His73 Cα atom (*Figure 2A*) but also the one to the $N^1$ atom of His73me (*Figure 2B*). In the substrate-bound structure, the imidazole ring of His73 stacks with the aromatic ring of Tyr313, with the $N^3$ atom of His73 in proximity to the hydroxyl group of Tyr313 (*Figure 2A*). The hydroxyl group of Tyr313 partially neutralizes the charge of $N^3$, as is verified by the fact that mutating Tyr313 to phenylalanine diminishes binding affinity 17-fold (Kd = 3.0 µM) (*Table 1*). A similar favorable charge–charge interaction has also been observed in the structure of Gemin5 bound to $m^7G$ base (*Xu et al., 2016*).

Given that the β-actin peptide interacts with SETD3 through the sequences that flank His73, we performed β-actin mutagenesis and ITC studies to corroborate the roles of Tyr69, Ile71, Asp80, Asp81, and Met82. The Tyr69, Ile71 and Met82 of β-actin are involved in binding to SETD3 via hydrophobic interactions, and the replacement of any of them with alanine leads to a 5–90 fold reduction in the SETD3 binding affinity (*Table 1*). Consistent with the fact that Asp80 and Asp81 of β-actin both form hydrogen bonds with SETD3, D80A and D81A cause a 20- and 11-fold decrease in SETD3 binding affinity, respectively (*Table 1*). The interactions between Asp81 and Met82 of β-actin and SETD3 also explain why truncation of the Asp81–His88 of the peptide reduces the binding to SETD3 ~17 fold (*Figure 2—figure supplement 1*, *Table 1*). Taken together, the flanking sequences of His73 on the β-actin peptide, rather than His73 itself, are critical for binding to SETD3, suggesting that the recognition of substrate by SETD3 is highly sequence-selective.

To obtain further insights into how substrate recognition by SETD3 affects subsequent histidine methylation, we used mass spectrometry to compare the activity of the above-mentioned SETD3 mutants with that of the wildtype protein in the presence of AdoMet. The mass spectrometry data indicate that the peaks of products generated by the four SETD3 mutants (R215A, N256A, Y313F, and R316A) were much lower than that observed for wildtype SETD3 (*Figure 1—figure supplement 1E*), suggesting that these mutants displayed weaker histidine methylation activity.

## Activities of the SETD3 enzyme and its mutant variants toward the β-actin protein

To verify the identification of the key catalytic residues of SETD3, and to analyze the roles of substrate recognition residues in catalysis, we determined the kinetic properties of wildtype and mutant variants of purified recombinant SETD3 in the presence of homogenous full-length recombinant human β-actin. All enzyme assays were performed in the presence of *E. coli* AdoHcy nucleosidase and *Bacillus subtilis* adenine deaminase to prevent the accumulation of AdoHcy in the reaction

**Table 3.** Kinetic properties of wildtype SETD3 and its mutants.
Kinetic parameters were determined by using purified recombinant N-terminal His$_6$-tagged SETD3 proteins.

| SETD3 | Substrate | Kinetic parameters | | | |
|---|---|---|---|---|---|
| | | $V_{max}$ | $K_M$ | $k_{cat}$ | $k_{cat}/K_M$ |
| | | $nmol\ min^{-1}\ mg^{-1}$ | $\mu M$ | $min^{-1}$ | $min^{-1}\ \mu M^{-1}$ |
| Wildtype | β-actin | 13.550 ± 0.364 | 0.502 ± 0.041 | 0.809 | 1.612 |
| R75A | | 0.037 ± 0.002 | 0.565 ± 0.071 | 0.002 | 0.004 |
| R215A | | 3.553 ± 0.068 | 1.087 ± 0.056 | 0.212 | 0.195 |
| N256A | | 0.512 ± 0.019 | 0.449 ± 0.053 | 0.031 | 0.069 |
| N256D | | 0.973 ± 0.017 | 0.305 ± 0.0196 | 0.058 | 0.190 |
| N256Q | | 10.860 ± 0.172 | 0.784 ± 0.035 | 0.649 | 0.828 |
| N278A | | 0.019 ± 0.000 | 0.047 ± 0.009 | 0.001 | 0.021 |
| Y313F | | 0.367 ± 0.005 | 0.581 ± 0.023 | 0.022 | 0.038 |
| R316A | | 2.903 ± 0.0477 | 1.115 ± 0.049 | 0.173 | 0.155 |
| Wildtype | AdoMet | 11.260 ± 0.466 | 0.111 ± 0.020 | 0.673 | 6.063 |
| R75A | | 0.265 ± 0.072 | 3.601 ± 1.363 | 0.016 | 0.004 |
| R215A | | 3.666 ± 0.078 | 0.165 ± 0.013 | 0.219 | 1.327 |
| N256A | | 1.136 ± 0.038 | 0.686 ± 0.052 | 0.068 | 0.099 |
| N256D | | 0.671 ± 0.021 | 0.165 ± 0.019 | 0.040 | 0.242 |
| N256Q | | 6.368 ± 0.266 | 0.103 ± 0.019 | 0.380 | 3.689 |
| N278A | | 0.018 ± 0.001 | 0.350 ± 0.042 | 0.001 | 0.003 |
| Y313F | | 1.623 ± 0.348 | 6.543 ± 1.736 | 0.097 | 0.015 |
| R316A | | 4.099 ± 0.054 | 0.149 ± 0.008 | 0.244 | 1.638 |

Determinations for *S*-adenosyl-L-methionine (AdoMet) were performed with the SETD3 preparations (0.05–5.00 µg protein, 8.38–838 nM), which were incubated for 10 min at 37°C in a reaction mixture containing 5 µM recombinant human β-actin and variable concentrations of [$^1$H+$^3$H] AdoMet (≈330 × 10$^3$ cpm). The measurements for β-actin were done following a 10 min incubation of SETD3 in the presence of a 0.8 µM concentration of [$^1$H+$^3$H] AdoMet (80 pmol, 300–700 × 10$^3$ cpm). In all experiments, the reaction mixture contained the homogenous recombinant AdoHcy nucleosidase (1.6 µg protein, 600 nM, *E. coli*) and adenine deaminase (3.9 µg protein, 600 nM, *B. subtilis*) to prevent *S*-adenosyl-L-homocysteine (AdoHcy) accumulation. Values are the means of three separate experiments. The S.E. values are also given.
DOI: https://doi.org/10.7554/eLife.43676.017

The following source data is available for Table 3:
**Source data 1.** Determination of the kinetic parameters of SETD3-catalyzed methylation of actin (for β-actin as the substrate).
DOI: https://doi.org/10.7554/eLife.43676.018

**Source data 2.** Determination of kinetic parameters of SETD3-catalyzed methylation of actin (for AdoMet as the substrate).
DOI: https://doi.org/10.7554/eLife.43676.019

mixture (*Dorgan et al., 2006*). As shown in *Table 3*, the wildtype SETD3 has a very high affinity toward both AdoMet ($K_M$ ≈ 0.1 µM) and β-actin ($K_M$ ≈ 0.5 µM), although it appeared to be a sluggish catalyst, with $k_{cat}$ equal to about 0.7–0.8 min$^{-1}$. All mutant enzymes exhibited a reduced catalytic efficiency as indicated by $k_{cat}/K_M$ ratios, which were 2–400-fold and 1.6–2000-fold for β-actin and AdoMet, respectively (*Table 1*), confirming the importance of the examined amino-acid residues for efficient catalysis.

Arg75 and Asn278 are both AdoMet-binding residues and the mutation of either of these to alanine severely impaired the $k_{cat}$ of the enzyme toward β-actin and AdoMet (c.f. *Table 3*). Interestingly, although N256A and Y313F weakened binding to the β-actin peptide (by ~12 and~17 fold, respectively (*Table 1*)), both mutants displayed $K_M$ values (0.449 µM and 0.581 µM, respectively) comparable to that of wildtype SETD3 (0.502 µM), probably because their $k_{cat}$ values are much lower (~26- and ~36-fold, respectively) than that of wildtype SETD3 (c.f. *Table 3*). For AdoMet, N256A and Y313F both have a higher $K_M$ (~6- and ~58-fold, respectively) and lower $k_{cat}$ (~9- and ~6-fold, respectively) than the wildtype enzyme. We also examined the kinetic properties of N256D and N256Q, and found that, for both β-actin and AdoMet, the kinetic parameters of N256Q are comparable to those of wildtype SETD3, while N256D behaves similarly to N256A (*Table 3*). Taking into account the fact that Asn256 and Tyr313 of SETD3 interact mainly with His73 of β-actin, and not with AdoMet (*Figure 2A–2B*), these results may imply that Asn256 and Tyr313 of SETD3 plausibly bind to His73 of β-actin by favorably orienting its imidazole ring to facilitate the efficient transfer of the methyl group from AdoMet to His73. The consequence of the N256A and Y313F mutations is thus a decrease in the rate of 'successful' methylation events, which could be partially overcome by an increase in AdoMet concentration.

In contrast to Asn256 and Tyr313, the Arg215 and Arg316 residues of SETD3 contact with the flanking sequences of His73. R215A and R316A, which impaired the binding of SETD3 to the β-actin peptide, also displayed higher $K_M$ values (~2-fold for both) and lower $k_{cat}$ values (~3.5- and~4.5-fold, respectively) for the β-actin protein (*Table 3*), confirming that the SETD3 residues that are involved in β-actin peptide recognition are also important for efficient His73 methylation in the context of the full-length β-actin.

We further investigated the pH-dependent activity of SETD3 toward full-length β-actin and found that SETD3 catalyzes the methylation of β-actin at a wide range of pH (6.0–9.0) environments, with its activity increasing with pH and reaching its maximum at pH 9.0 (*Figure 3*). The detectable activity of SETD3 at neutral pH is consistent with the fact that histidine deprotonation occurs at neutral pH, which facilitates the transfer of the methyl group from AdoMet to histidine. The increasing rate of the reaction with increasing pH to pH 9.0 could suggest, however, that the imidazolium pKa is perturbed (elevated) in the active site due to the active site environment or that proton transfer from the imidazole/imidazolium may not contribute to the rate-limiting step.

## Discussion

### Catalytic mechanism of SETD3

As the major component of microfilaments, actin participates in diverse cellular processes. Different types of PTM have been identified in actin, and these modifications play important roles in mediating actin functions in vivo (*Terman and Kashina, 2013*). N$^3$-methylhistidine was identified in β-actin decades ago (*Johnson et al., 1967*) and was postulated to mediate the polymerization and hydrolysis of actin filaments (*Kabsch et al., 1990*), but how the histidine methylation is incorporated remains enigmatic. Our identification of SETD3 as a histidine methyltransferase that catalyzes the His73 methylation of β-actin answers this unsolved question (*Kwiatkowski et al., 2018*).

Although SETD3 contains a canonical SET domain and its AdoHcy-binding mode is similar to those observed for other SET domain lysine methyltransferases, the predicted SET-domain fold of SETD3 does not help to disclose the molecular mechanisms of the enzyme's substrate recognition and histidine methylation activities, because the SET domain itself, as a short motif of ~110 aa, always requires the iSET and C-SET domains to complete its function (*Chang et al., 2011*; *Trievel et al., 2002*). Our solved substrate- and product-bound structures of SETD3 provide an insight into the pre-and post- methyl transfer states of this enzyme as it catalyses the histidine

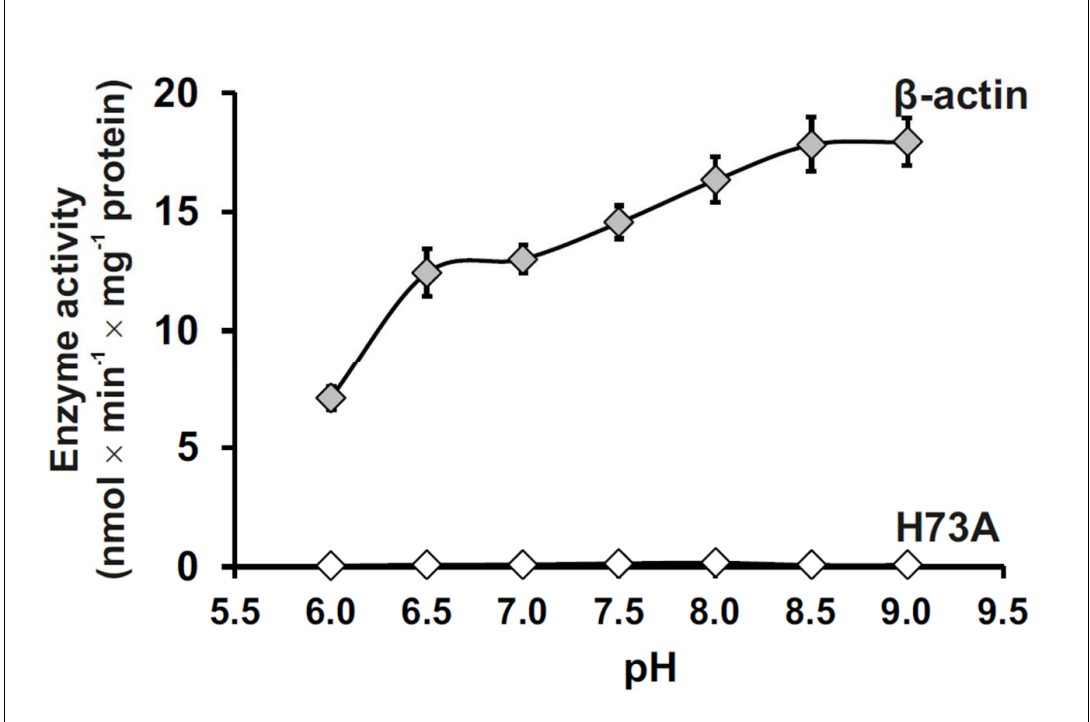

**Figure 3.** Effect of pH on SETD3 enzyme activity. The pH-dependence of human SETD3 was determined with the use of a purified recombinant N-terminal His$_6$-tagged SETD3 protein (aa 2–502). The enzyme preparation (0.045 µg protein) was incubated at 37°C for 10 min in the presence of 0.8 µM [$^1$H+$^3$H] AdoMet (80 pmol, ≈350 × 10$^3$ cpm) and either 5 µM (500 pmol, 22.34 µg) purified recombinant human β-actin or its mutated form (H73A), with the latter serving as a negative control. In all experiments, the reaction mixture contained the homogenous recombinant AdoHcy nucleosidase (1.6 mg protein, 600 nM, *E. coli*) and adenine deaminase (3.9 mg protein, 600 nM, *B. subtilis*) to prevent AdoHcy accumulation. The reaction was stopped and the proteins present in the assay mixture were precipitated by adding 10% trichloroacetic acid. This allowed for the separation and specific measurement of the radioactivity incorporated in the protein pellet (representing the extent of actin methylation) from the total radioactivity present in the assay mixture. Values are the means ± S.E. (error bars) of three separate experiments. If no error bar is visible, it is shorter than the height of the symbol.

DOI: https://doi.org/10.7554/eLife.43676.020

The following source data is available for figure 3:

**Source data 1.** Radiochemical measurements of SETD3-dependent methylation of either human recombinant β-actin or its mutated form (H73A) in the presence of increasing pH values of the reaction mixture.

DOI: https://doi.org/10.7554/eLife.43676.021

methylation of its substrate. For the first time, these SETD3 structures reveal the molecular mechanism by which a SET domain protein acts as the actin His73-specific methyltransferase.

Our structural analyses, assisted by mutagenesis and biochemical experiments, suggest that the recognition of β-actin and its subsequent catalysis by SETD3 are different from those observed for other SET-domain-containing lysine methyltransferases in the following respects: (i) the histidine recognition pocket of SETD3 is shallower than that of other SET lysine methyltransferases, and key residues of SETD3 that form hydrogen bonds with the His73 imidazole ring are not conserved in other SET proteins (*Figure 2A–2B*, *Figure 1—figure supplement 2B*); (ii) in two solved SETD3 complex structures, the methylated histidine rotates its side chain by 90° (*Figure 2C*); and (iii) the recognition of β-actin by SETD3 is highly dependent on the sequences flanking His73 of β-actin (*Figure 2—figure supplement 1*), consistent with the high binding affinity between SETD3 and a β-actin H73A peptide (66–88) (*Figure 1B*). Although N$^3$-methylhistidine is also found in other proteins, such as mammalian myosin (*Johnson et al., 1967*) and *Saccharomyces cerevisiae* Rpl3 (*Webb et al., 2010*), the sequence preference suggests that SETD3 probably works only on His73 of β-actin, as well as on the corresponding histidines in α- and γ-actins. One implication is that no SETD3 ortholog has been found in *S. cerevisiae*.

Previously, we identified *Carnosine N-methyltransferase 1* (*CARNMT1*) as the gene that encodes carnosine *N*-methyltransferase in mammals (*Drozak et al., 2015*). This enzyme catalyzes $N^1$-methylation of the histidine imidazole ring of carnosine (β-alanyl-L-histidine), an abundant dipeptide in the skeletal muscle of vertebrates. Recently, the structures of CARNMT1 bound to analogs of $N^1$-methylhistidine were also reported (*Cao et al., 2018*), allowing us to compare $N^3$-histidine methylation by SETD3 with $N^1$-histidine methylation by CARNMT1 by studying the two complexes (*Figure 2A and E*). We found that SETD3 and CARNMT1 not only adopt different folds and AdoHcy-binding modes, but also display distinct histidine recognition and catalysis mechanisms. Asp316 of CARNMT1 forms two hydrogen bonds with the imidazole ring of the histidine to facilitate its deprotonation (*Figure 2E*) (*Cao et al., 2018*). In the SETD3–substrate complex, His73 stacks with Tyr313 and is hydrogen-bonded to the main chain carbonyl group of Asp275 (*Figure 2A*), which orients the His73 imidazole ring for efficient catalysis. As determined by the mutagenesis, ITC binding experiments and enzymatic assays (*Table 1*, *Table 3*), Tyr313 of SETD3 plays important roles in substrate recognition and in subsequent histidine methylation. Finally, the 90° rotation of the His73me imidazole ring was not found in CARNMT1-mediated histidine methylation.

## Implications of conformational changes in actin upon binding to SETD3

In natively purified rabbit β-actin, His73 exists in the methylated form and it is localized close to the phosphate groups of ATP binding to actin (*Kabsch et al., 1990*). Although our complexes contain only a fragment of β-actin, they could still provide clues about the role of SETD3 in mediating actin functions. The residues 66–84 of β-actin consist of three β strands followed by an α-helix; however, the β-actin peptide adopts an extended conformation upon binding to SETD3 (*Figure 2—figure supplement 3*), suggesting that β-actin probably endures local structural remodeling during methylation. Given that SETD3 is found to be co-purified with other factors, such as the peptide release factor ERF3A (*Kwiatkowski et al., 2018*), SETD3 might require the assistance of other protein(s) to achieve the methylation of actin His73.

## Conclusions

In the current investigation, we have identified the molecular mechanism of β-actin methylation by SETD3 and have shown that the enzyme displays a very high substrate specificity toward β-actin His73. The overexpression of SETD3 was recently reported to be associated with malignant cancer (*Cheng et al., 2017*), so the high-resolution complex structures of SETD3 might be helpful in providing a better understanding of the role of actin histidine methylation in human cancer, as well as in providing a structural basis for the design of specific SETD3 inhibitors in the near future.

During the peer review process for our manuscript, a manuscript was published that reveals the function of mammalian SETD3 in preventing primary dystocia and that presents the structures of SETD3 bound to an unmodified actin peptide (aa 66–80) (*Wilkinson et al., 2019*). The structural findings reported in this manuscript are similar to ours, except that our complex structures contain a longer actin fragment (aa 66–88) and that the histidine is methylated in one of our complexes.

## Materials and methods

**Key resources table**

| Reagent type (species) or resource | Designation | Source or reference | Identifiers | Additional information |
|---|---|---|---|---|
| ene (*E. coli*) | AdoHcy nucleosidase | NA | NCBI: NC_000913.3 | |
| Gene (*B. subtilis*) | adenine deaminase | NA | NCBI: NC_000964.3 | |
| Gene (*Homo sapiens*) | β-actin | NA | GenBank: NM_001101.4 | |
| Gene (*H. sapiens*) | SETD3 | NA | NCBI: NM_032233.2 | |
| Recombinant DNA reagent | pCOLD I (plasmid) | Takara Bio | 3361 | |
| Genetic reagent (*E. coli*) | pCOLD I/β-actin | PMID: 20851184 | | Kind gift from M. Tamura, Ehime University |

*Continued on next page*

*Continued*

| Reagent type (species) or resource | Designation | Source or reference | Identifiers | Additional information |
|---|---|---|---|---|
| Genetic reagent (*E. coli*) | pCOLD I/AdoHcy nucleosidase | PMID: 30526847 | | Overexpression of the recombinant *E. coli* AdoHcy nucleosidase in *E. coli* BL21 (DE3) |
| Genetic reagent (*E. coli*) | pCOLD I/adenine deaminase | PMID: 30526847 | | Overexpression of the recombinant *B. subtilis* adenine deaminase in *E. coli* BL21 (DE3) |
| Strain, strain background (*B. subtilis*) | *B. subtilis* | Sigma-Aldrich | ATCC:6633 | |
| Strain, strain background (*E. coli*) | *E. coli* BL21 (DE3) | Agilent Technologies | 200131 | |
| Chemical compound, drug | S-[methyl-$^3$H]adenosyl-L-methionine; [$^3$H]SAM | PerkinElmer | NET155V250UC; NET155V001MC | |
| Software, algorithm | GraphPad Prism | GraphPad Software | RRID:SCR_002798 | |
| pET28-MHL | Vector | Addgene | 26096 (Genbank accession number: EF456735) | Expression of SETD3 and its mutants |
| Origin 6.1, Origin 7.0 | Software | OriginLab | http://www.originlab.com/ | For ITC curve fitting and calculation of Kd values |
| HKL2000 | Software | PMID: 27754618 | http://www.hkl-xray.com/ | Processing the crystal structure data |
| Phaser | Software | PMID: 19461840 | http://www.phaser.cimr.cam.ac.uk/ | Molecular replacement |
| Phenix | Software | PMID: 20124702 | http://www.phenix-online.org/ | Structure refinement |
| Coot | Software | PMID: 15572765 | http://www2.mrc-lmb.cam.ac.uk/Personal/pemsley/coot/ | Structure refinement |
| Pymol | Software | DeLano Scientific LLC | http://www.pymol.org/ | Making structure figures |
| Molprobity | Software | PMID: 20057044 | http://molprobity.biochem.duke.edu/ | Structure validation |

## Cloning, protein expression and purification of SETD3

Genes encoding the core region of SETD3 (aa 2–502) were synthesized by Sangon Biotech (Shanghai) and cloned into pET28-MHL (Genbank accession number: EF456735). Then, the plasmid was transformed into *E. coli* BL21 (DE3) and the recombinant protein was overexpressed at 16°C for 20 hr in the presence of 0.5 mM isopropyl β-D-1-thiogalactopyranoside (IPTG). Recombinant SETD3 proteins were purified with a fast flow Ni-NTA column (GE Healthcare). N-terminal 6 × His tags of recombinant proteins were removed with Tobacco Etch Virus (TEV) protease. Gel filtration and ion-exchange were employed for further purification. Purified SETD3 proteins, with their purities exceeding 90% as assessed by SDS-PAGE, were concentrated to 15 mg/ml and stored at −80°C before further use. The SETD3 mutants were constructed by conventional PCR using a MutanBEST kit (TaKaRa) and further verified by DNA sequencing. The SETD3 mutants were expressed and purified in the same way as the wildtype protein.

For kinetic studies, the enzyme and its mutant forms were produced as fusion proteins with the N-terminal 6 × His tag and purified by Ni-NTA affinity chromatography. Briefly, the *E. coli* cell paste was resuspended in 11 ml lysis buffer consisting of 25 mM Hepes (pH 7.5), 300 mM NaCl, 10 mM KCl, 1 mM dithiothreitol (DTT), 2 mM MgCl$_2$, 1 mM phenylmethanesulfonylfluoride (PMSF), 0.25 mg/ml hen egg white lysozyme (BioShop, Canada) and 250 U Viscolase (A and A Biotechnology, Poland). The cells were lysed by freezing in liquid nitrogen and, after thawing and vortexing, the extracts were centrifuged at 4°C (20,000 × *g* for 20 min). For the purification, the supernatant of *E. coli* lysate (11 ml) was diluted 3-fold with buffer A (50 mM Tris-HCl (pH 7.4), 400 mM NaCl, 10 mM KCl and 1

mM DTT) and applied onto a HisTrap FF crude column (1 ml) equilibrated with the 90% buffer A and 10% buffer B (30 mM imidazole). The column was then washed with 12 ml 90% buffer A and 10% buffer B, and the retained proteins were eluted with a stepwise gradient of imidazole (5 ml of 60 mM, 5 ml of 150 mM and 5 ml of 300 mM) in buffer A. The recombinant proteins eluted at the highest concentration of imidazole were further processed. The enzyme preparation was desalted onto PD-10 columns equilibrated with 20 mM Tris-HCl (pH 7.2), 50 mM KCl, 1 mM DTT and 6% sucrose. The yield of recombinant proteins ranged from 1.6 mg to 0.1 mg per 200 ml of culture for the wild-type and the N278A mutant, respectively. The purified proteins were aliquoted and stored at –70°C.

## Overexpression and purification of the recombinant β-actin inclusion-body protein

Recombinant human β-actin (ACTB, GenBank: NM_001101.4) was prepared as described previously (*Kwiatkowski et al., 2018*). The plasmid pCOLD I that encodes the human protein was a kind gift from Dr. Minoru Tamura (Ehime University, Japan) and was prepared as described by *Tamura (2018)*.

For β-actin production, *E. coli* BL21(DE3) (Agilent, USA) cells were transformed with the DNA construct and cultured in 500 ml of LB broth (with 100 μg/ml ampicillin) at 37°C and 200 rpm until an $OD_{600}$ of 0.5 was reached. Protein production was induced by cold-shock (20 min on ice) and IPTG addition to a final concentration of 0.2 mM. β-Actin production was carried out for 20 hr at 15°C, 200 rpm, and harvested by centrifugation (6,000 × $g$ for 10 min). The cell paste was resuspended in 27.5 ml lysis buffer consisting of 20 mM Hepes (pH 7.5), 1 mM DTT, 1 mM ADP, 0.5 mM PMSF, 2 μg/ml leupeptin, 2 μg/ml antipain, 0.2 mg/ml hen egg white lysozyme (BioShop), and 1,000 U Visco-lase (A and A Biotechnology, Poland). The cells were lysed by freezing in liquid nitrogen and, after thawing and vortexing, the extracts were centrifuged at 4°C (20,000 × $g$ for 30 min).

The pellet containing inclusion bodies was completely resuspended in buffer A (20 mM Hepes (pH 7.5), 2 M urea, 0.5 M NaCl, 5 mM DTT, 2 mM EDTA) with the use of a Potter-Elvehjem homogenizer and centrifuged at 4°C (20,000 × $g$ for 10 min). The resulting pellet was then subjected to a further two rounds of sequential wash in buffer B (20 mM Hepes (pH 7.5), 0.5 M NaCl, 5 mM DTT, 2 mM EDTA) and buffer C (20 mM Hepes (pH 7.5), 0.5 M NaCl). The washed inclusion bodies were finally solubilized in loading buffer (20 mM Tris-HCl (pH 7.5), 6 M guanidine HCl, 0.5 M NaCl, 10 mM imidazole) and applied on a HisTrap FF column (5 ml) equilibrated with the same buffer.

The column was washed with 20 ml of loading buffer and the bound protein was refolded with the use of a linear gradient of 6–0 M guanidine HCl in loading buffer (40 ml for 20 min). Next, the column was washed with 15 ml of loading buffer without guanidine HCl and the retained proteins were eluted with a stepwise gradient of imidazole (25 ml of 40 mM, 25 ml of 60 mM and 20 ml of 500 mM). The recombinant proteins were eluted with 500 mM imidazole in homogeneous form as confirmed by SDS-PAGE (not shown). The β-actin preparation (10 ml) was immediately dialyzed against 400 ml of the buffer consisting of 20 mM Tris-HCl (pH 7.5), 1 mM DTT, 6% sucrose, 2 μg/ml leupeptin and 2 μg/ml antipain. The dialysis buffer was exchanged three times with the following dialysis times: 2, 2 and 12 hr. The purified β-actin was stored at −70°C.

## Overexpression and purification of the recombinant AdoHcy nucleosidase and adenine deaminase

*Escherichia coli* DNA was extracted by heating 50 μl of over-night cultured BL21(DE3) cells at 95°C for 15 min, whereas *B. subtilis* (ATCC 6633, Sigma-Aldrich) genomic DNA was purified from 100 mg of bacterial cells with the use of TriPure reagent (Roche, Swizerland) according to the manufacturer's instructions.

The open reading frames encoding *E. coli* AdoHcy nucleosidase and *B. subtilis* adenine deaminase (NCBI Reference Sequence: NC_000913.3 and NC_000964.3, respectively) were PCR-amplified using either Pfu DNA polymerase alone or a mixture of Taq:Pfu polymerases (1:0.2), respectively, in the presence of 1 M betaine. The AdoHcy nucleosidase ORF was amplified using a 50-nucleotide primer containing the initiator codon preceded by an NdeI site and a 30-nucleotide primer with a HindIII site, whereas adenine deaminase DNA was amplified using a 50-nucleotide primer with the initiator codon preceded by a KpnI site and a 30-nucleotide primer with a BamHI site (for primer sequences, see *Kwiatkowski et al., 2018*). Amplified DNA products of expected size were digested

with the appropriate restriction enzymes, cloned into the pCOLD I expression vector (pCOLD I/ AdoHcy nucleosidase and pCOLD I/adenine deaminase) and verified by DNA sequencing (Macrogen, The Netherlands).

For protein production, *E. coli* BL21(DE3) cells were transformed with the appropriate DNA construct and a single colony was selected to start an over-night pre-culture. 100 mL of LB broth (with 100 mg/mL ampicillin) was inoculated with 10 ml of the pre-culture and incubated at 37°C and 200 rpm until an OD600 of 0.6 was reached. The culture was placed on ice for 20 min (cold-shock) and IPTG was added to a final concentration of 0.25 mM to induce protein expression. Cells were incubated for 16 hr at 13°C, 200 rpm, and harvested by centrifugation (6,000 × *g* for 10 min). The cell pellet was resuspended in 10 ml lysis buffer consisting of 25 mM Hepes (pH 7.5), 300 mM NaCl, 50 mM KCl, 1 mM DTT, 2 mM MgCl2, 1 mM PMSF, 5 mg/ml leupeptin and 5 mg/ml antipain, together with 0.25 mg/ml hen egg white lysozyme (BioShop) and 25 U DNase I (Roche). The cells were lysed by freezing in liquid nitrogen and, after thawing and vortexing, the extracts were centrifuged at 4°C (20,000 × *g* for 30 min).

For protein purification, the supernatant of the *E. coli* lysate (10 ml) was diluted 3-fold with buffer A (50 mM Tris-HCl (pH 7.2), 400 mM NaCl, 10 mM KCl, 30 mM imidazole, 1 mM DTT, 3 mg/ml leupeptin and 3 mg/ml antipain) and applied onto a HisTrap FF column (5 ml) equilibrated with the same buffer. The column was washed with 20–30 ml buffer A and the retained protein was eluted with a stepwise gradient of imidazole (25 ml of 60 mM, 20 ml of 150 mM and 20 ml of 300 mM) in buffer A. The recombinant proteins were eluted with 150–300 mM imidazole in homogeneous form, as confirmed by SDS-PAGE (not shown). The enzyme preparations were desalted onto PD-10 columns equilibrated with 20 mM Tris-HCl (pH 7.2), 50 mM KCl, 1 mM DTT, 6% sucrose, 2 μg/ml leupeptin and 2 μg/ml antipain. The yield of recombinant proteins was 1.2 mg and 3.1 mg of homogenous adenine deaminase and AdoHcy nucleosidase, respectively, per 200 ml of *E. coli* culture. The purified enzymes were aliquoted and stored at −70°C.

## Isothermal titration calorimetry (ITC)

All peptides were synthesized by GL Biochem (Shanghai) Ltd. and were dissolved in water as a stock of 5–12 mM, with the pH of stock solutions adjusted to pH 7.5. Peptides and concentrated proteins were diluted with ITC buffer (20 mM Tris, pH 7.5 and 150 mM NaCl). ITC experiments were performed by titrating 2 μl of peptide (1.2–1.5 mM) into cells containing 50 μM proteins on MicroCal (Malvern Panalytical, UK) at 25°C, with a spacing time of 160 s and a reference power of 10 μCal/s. Control experiments were performed by injection of peptides into buffer. Binding isotherms were plotted, analyzed and fitted in a one-site binding model by MicroCal PEAQ-ITC Analysis Software (Malvern Panalytical, UK) after subtraction of the respective controls. The dissociation constants (Kds) were determined from a minimum of two experiments (mean ± SD). The ITC binding curves are shown in *Supplementary file 1*.

## Crystallization, data collection and structure determination

All crystals were grown using the sitting drop vapor diffusion method at 18°C. For crystallization of SETD3 with methylated peptide, SETD3 (12 mg/ml) was pre-incubated with synthesized β-actin peptides (aa 66–88) (GL Biochem Ltd.) and AdoMet at a molar ratio of 1:3:4, and mixed with the crystallization buffer containing 0.1 M sodium cacodylate trihydrate (pH 6.5), 0.2 M magnesium acetate tetrahydrate, and 20% v/v polyethylene glycol 8000. For the crystallization of SETD3 with unmodified peptide, SETD3, at the concentration of 12 mg/ml, was pre-incubated with synthesized β-actin peptides (aa 66–88) and AdoHcy at a molar ratio of 1:3:4, and mixed with the crystallization buffer containing 0.1M HEPES sodium (pH 7.5), 2% v/v polyethylene glycol 400, and 2.0 M ammonium sulfate. Before flash-freezing crystals in liquid nitrogen, all crystals were soaked in a cryo-protectant consisting of 90% reservoir solution plus 10% glycerol. The diffraction data were collected on BL17U1 at the Shanghai Synchrotron Facility (*Wang et al., 2016*) (SSRF). Data sets were collected at 0.9789 Å or 0.9789 Å, and were processed using the HKL2000 program (*Otwinowski and Minor, 1997*).

The initial structures of the SETD3-actin complexes were solved by molecular replacement using Phaser (*McCoy et al., 2007*) with a previously solved SETD3 structure (PDB: 3SMT) as the search model. Then, all of the models were refined manually and built with Coot (*Emsley and Cowtan,*

*2004*). The final structures were further refined by PHENIX (*Adams et al., 2010*). The statistics for data collection and refinement are summarized in *Table 2*.

## Mass spectrometry

Reversed-phase microcapillary/tandem mass spectrometry (LC/MS/MS) was performed using an Easy-nLC nanoflow HPLC (Proxeon Biosciences) with a self-packed 75 µm × 15 cm C18 column connected to a QE-Plus (Thermo Scientific) in data-dependent acquisition and positive ion mode at 300 nL/min. Passing MS/MS spectra were manually inspected to ensure that all b-and y-fragment ions aligned with the assigned sequence and modification sites. A 25 ul reaction mixture contained 2 µM SETD3 or SETD3 mutants (final concentration) and 20 µM peptide (final concentration) in a buffer containing 10 mM Tris-HCl, (pH 7.5), 20 mM NaCl and 10 µM AdoMet. The reaction was incubated at 37°C for 2 hr before being quenched (at 70°C for 10–15 mins). Then, reactions were analyzed by LC/MS/MS and Proteomics Browser software, with the relative abundances of substrate and product reflecting the methylation activities of proteins.

## Radiochemical assay of SETD3 activity

Enzyme activity was determined by measuring the incorporation of the [$^3$H]methyl group from *S*-[methyl-$^3$H]adenosyl-L-methionine ([$^3$H]AdoMet) into homogenous recombinant human (mammalian) β-actin or its mutated form in which histidine 73 was replaced by an alanine residue (H73A). The standard incubation mixture (0.06–0.11 ml) contained 25 mM Tris-HCl (pH 7.2), 10 mM KCl, 1 mM DTT and various concentrations of protein substrate and [$^1$H+$^3$H] AdoMet ($\approx$300–700 × 10$^3$ cpm), as indicated in the legends to the figures and tables. When the effect of pH on recombinant enzyme activity was investigated, 25 mM Tris-HCl (pH 7.2) was replaced by 40 mM MES (pH 6.0 or 6.5), 40 mM Hepes (pH 7.0 and 7.5), or 40 mM Tris (pH 8.0, 8.5 or 9.0).

The incubation mixture was supplemented with recombinant AdoHcy nucleosidase and adenine deaminase to prevent AdoHcy accumulation, as indicated in the legends to the figures and tables. The reaction was started by the addition of an enzyme preparation and carried out at 37°C for 10 min. Protein methylation was linear for at least 15 min under all conditions studied. By analogy to assays of non-ribosomal peptide synthetase activity (*Drozak et al., 2014*; *Richardt et al., 2003*), the incubation was stopped by the addition of 0.05–0.11 ml of the reaction mixture to 0.025 ml of bovine serum albumin (BSA) (1 mg) and 0.8 ml of ice-cold 10% (w/v) trichloroacetic acid (TCA). After 10 min on ice, the precipitate was pelleted and washed twice with ice-cold 10% TCA. The pellet was finally dissolved in pure formic acid.

## Calculations

$V_{max}$, $K_M$ and $k_{cat}$ were calculated for the methyltransferase activity of the studied enzymes with Prism 8.0 (GraphPad Software, La Jolla, USA) using a nonlinear regression.

## Accession numbers

The coordinates and structure factors of AdoHcy-bound SETD3 with unmodified β-actin peptide and methylated β-actin peptide have been deposited into the Protein Data Bank (PDB) with accession numbers 6ICV and 6ICT, respectively.

## Acknowledgements

We are grateful to the staff members at beam lines BL17U1, BL18U1 and BL19U1 at the Shanghai Synchrotron Radiation Facility for their assistance in data collection. This work was supported by the National Natural Science Foundation of China Grants 31770806 (to CX), 31570737 (to CX), 31500601 (to SL), and 31501093 (to HY). JD is supported by Narodowe Centrum Nauki (Opus Grant UMO-2017/27/B/NZ1/00161). HY is supported by a China Postdoctoral Science Foundation Grant (No. 2015M580547). CX is supported by the Major/Innovative Program of the Development Foundation of the Hefei Center for Physical Science and Technology (2018CXFX007), the 'Thousand Young Talent program' and the Fundamental Research Funds for the Central Universities (WK2070080001)." The Structural Genomics Consortium is a registered charity (no. 1097737) that receives funds from: AbbVie; Bayer Pharma AG; BoehringerIngelheim; the Canada Foundation for

Innovation; the Eshelman Institute for Innovation; Genome Canada through the Ontario Genomics Institute; the Innovative Medicines Initiative (European Union/European Federation of Pharmaceutical Industries and Associations; Unrestricted Leveraging of Targets for Research Advancement and Drug Discovery [ULTRA-DD] grant no. 115766); Janssen, Merck, and Company; Novartis Pharma AG; the Ontario Ministry of Economic Development and Innovation; Pfizer; the São Paulo Research Foundation (FAPESP); Takeda; and the Wellcome Trust (to JM).

## Additional information

### Funding

| Funder | Grant reference number | Author |
| --- | --- | --- |
| National Natural Science Foundation of China | 31500601 | Shanhui Liao |
| National Natural Science Foundation of China | 31501093 | Huijuan Yu |
| China Postdoctoral Science Foundation | 2015M580547 | Huijuan Yu |
| Narodowe Centrum Nauki | UMO-2017/27/B/NZ1/00161 | Jakub Drozak |
| National Natural Science Foundation of China | 31570737 | Chao Xu |
| National Natural Science Foundation of China | 31770806 | Chao Xu |
| University of Science and Technology of China | 2018CXFX007 | Chao Xu |
| Fundamental Research Funds for the Central Universities | WK2070080001 | Chao Xu |

The funders had no role in study design, data collection and interpretation, or the decision to submit the work for publication.

### Author contributions
Qiong Guo, Software, Formal analysis, Validation, Investigation, Methodology; Shanhui Liao, Conceptualization, Software, Formal analysis, Supervision, Validation, Investigation, Methodology; Sebastian Kwiatkowski, Weronika Tomaka, Investigation, Methodology; Huijuan Yu, Formal analysis, Investigation, Methodology; Gao Wu, Jakub Drozak, Formal analysis, Supervision, Funding acquisition, Validation, Investigation, Methodology, Project administration, Writing—review and editing; Xiaoming Tu, Conceptualization, Investigation, Writing—review and editing; Jinrong Min, Conceptualization, Formal analysis, Supervision, Funding acquisition, Validation, Investigation, Methodology, Project administration, Writing—review and editing; Chao Xu, Conceptualization, Data curation, Formal analysis, Supervision, Funding acquisition, Validation, Investigation, Writing—original draft, Project administration, Writing—review and editing

### Author ORCIDs
Sebastian Kwiatkowski  http://orcid.org/0000-0003-4908-1633
Jakub Drozak  http://orcid.org/0000-0002-3601-3845
Chao Xu  http://orcid.org/0000-0003-0444-7080

### Decision letter and Author response
Decision letter https://doi.org/10.7554/eLife.43676.030
Author response https://doi.org/10.7554/eLife.43676.031

## Additional files

### Supplementary files

• Supplementary file 1. ITC binding curves for the binding measurements reported in *Table 1*.
DOI: https://doi.org/10.7554/eLife.43676.022

• Transparent reporting form
DOI: https://doi.org/10.7554/eLife.43676.023

### Data availability

Diffraction data have been deposited in PDB under the accession codes 6ICV and 6ICT.

The following datasets were generated:

| Author(s) | Year | Dataset title | Dataset URL | Database and Identifier |
|---|---|---|---|---|
| Qiong Guo, Shanhui Liao, Chao Xu | 2018 | Structure of SETD3 bound to SAH and unmodified actin | http://www.rcsb.org/structure/6ICV | Protein Data Bank, 6ICV |
| Qiong Guo, Shanhui Liao, Chao Xu | 2018 | Structure of SETD3 bound to SAH and methylated actin | http://www.rcsb.org/structure/6ICT | Protein Data Bank, 6ICT |

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
