## [Decision Letter]

Thank you for submitting your article "Structural insights into SETD3-mediated histidine methylation on β-actin" for consideration by *eLife*. Your article has been reviewed by Philip Cole as the Senior Editor and Reviewing Editor, and two reviewers. The following individuals involved in review of your submission have agreed to reveal their identity: Paul R Thompson (Reviewer #1).

The reviewers have discussed the reviews with one another and the Reviewing Editor has drafted this decision to help you prepare a revised submission.

Summary:

The manuscript by Guo et al. describes the molecular recognition of β-actin by SETD3. This methyltransferase has recently been reported to methylate the N3 atom in the side chain of H73 in β-actin. Crystal structures of SETD3 bound to a β-actin peptide illustrate extensive interactions between the enzyme and the residues surrounding H73 in the substrate. Mutations that disrupt these interactions diminish binding affinity and methylation of β-actin by SETD3. This work sheds new light on SETD3 recognition of β-actin and will likely be of broad interest to investigators in the cell biology, biochemistry, and PTMs field.

Essential revisions:

1) The catalytic mechanism shown in Figure 5D developed based in part on the structure is highly problematic. Among the major concerns: Y313 as a catalytic base shown as a phenoxide anion is not well justified and is improbable. Much more work would be needed to establish this including detailed pH rate profiles, substrate analogs etc. That Y313F mutation has only a moderate effect (<10-fold) on methyltransferase rate could readily be explained in other ways (substrate affinity, substrate orientation etc). It is quite possible that the imidazole side chain with its low pKa (ca. 7) would not need a catalytic base in any event. In addition, the proposed flipping of the N256 side chain (Supplementary figure S8) is inconsistent with the deposited structure of SETD3 (3SMT.pdb) wherein its carboxamide NH2 group hydrogen bonds with the R254 carbonyl oxygen. If the N256 carboxamide side chain were to flip, this would result in electrostatic repulsion between its oxygen and the R254 carbonyl oxygen. Additionally, N1 will be protonated whether the H73 imidazole group initially binds to SETD3 in a fully protonated state or an N3-deprontated state. Therefore, the N256 carboxamide oxygen should face toward the H73 imidazole ring and hydrogen bond to the N1 position. In sum, the N256 carboxamide ring-flipping discussion and Supplementary figure S8 should be removed from the paper and the N256 side chain in SETD3 should be modeled with proper hydrogen bonding in the deposited structures. Overall, our advice is to minimize discussion of catalysis and drastically simplify or remove the catalytic mechanism images unless and until much more extensive enzymologic work is performed.

2) The methyltransferase activity assay used here only measures relative rates. More detailed kinetic analysis is needed to discriminate binding effects from catalysis. If the substrate Km values are too high to determine, then this should be clearly indicated, and the limitations of the measurements stated more directly to readers. Furthermore, specific activities presented as V/E (velocity over enzyme concentration in min-1) should be shown.

3) The writing of the paper and its presentation needs major improvements. Key figures are placed in the supporting information whereas figures that minimally add value are included with the main figures. For example, Figure 2 could either be in the supporting info or be one panel of a multi-paneled figure. Figure 4 is of limited impact. The comparisons to LSMT and CARNMT1 could also be one panel of a multi-paneled figure. By contrast, the table that reports the effects of mutations on peptide binding is in the supporting info--this should be in the main section. Likewise, the electron density maps that depict the methylated histidine are in the supporting figures and this should also be a main figure. The activity data are in the supporting info and this should be in the main section.

The manuscript should be extensively edited for proper sentence structure, grammar and spelling to improve readability and comprehension. There are many convoluted sentences.

4) Because SETD3 recognizes a long β-actin peptide (residues 66-88), the authors should really test the activity and binding affinity of the enzyme using comparable length H3K4 and H3K36 peptides. The H3K4 (1-10) and H3K36 (31-44) peptides tested may have been too short for binding and methylation by SETD3. If this will be too time consuming, at the very least the authors should state this caveat in their interpretation of the comparative enzymologic studies.

---

## [Author Response]

Essential revisions:1) The catalytic mechanism shown in Figure 5D developed based in part on the structure is highly problematic. Among the major concerns: Y313 as a catalytic base shown as a phenoxide anion is not well justified and is improbable. Much more work would be needed to establish this including detailed pH rate profiles, substrate analogs etc. That Y313F mutation has only a moderate effect (<10-fold) on methyltransferase rate could readily be explained in other ways (substrate affinity, substrate orientation etc). It is quite possible that the imidazole side chain with its low pKa (ca. 7) would not need a catalytic base in any event. In addition, the proposed flipping of the N256 side chain (Supplementary figure S8) is inconsistent with the deposited structure of SETD3 (3SMT.pdb) wherein its carboxamide NH2 group hydrogen bonds with the R254 carbonyl oxygen. If the N256 carboxamide side chain were to flip, this would result in electrostatic repulsion between its oxygen and the R254 carbonyl oxygen. Additionally, N1 will be protonated whether the H73 imidazole group initially binds to SETD3 in a fully protonated state or an N3-deprontated state. Therefore, the N256 carboxamide oxygen should face toward the H73 imidazole ring and hydrogen bond to the N1 position. In sum, the N256 carboxamide ring-flipping discussion and Supplementary figure S8 should be removed from the paper and the N256 side chain in SETD3 should be modeled with proper hydrogen bonding in the deposited structures. Overall, our advice is to minimize discussion of catalysis and drastically simplify or remove the catalytic mechanism images unless and until much more extensive enzymologic work is performed.

We thank the referees for the constructive comments. We also admitted that Tyr313 is not absolutely required since Y313F mutant still catalyzes the methylation of the peptide, albeit with lower activity. We performed the pH profile of SETD3 activity towards full length recombinant β-actin and found SETD3 to display histidine methylation activity in a wide range of pH values, with a clear preference to the alkaline conditions (a new Figure 3 of the revised manuscript). That could be well explained by the fact that a majority of histidine deprotonates at high pH (pKa of the histidine imidazole moiety is approximately 6.0), which facilitates the transferring of methyl group from AdoMet to the substrate. Unlike lysine, histidine protonation/deprotonation occurs at neutral pH, the pH profiles implicates that a general base probably is not required for methylation. Thus, we agree with the Referee’s suggestion that SETD3 Tyr313 might play an important role in binding to and orienting the substrate (see response to comment 2).

We have now introduced a short paragraph into the Results section to provide information on the pH-dependence of SETD3 activity.

As for the orientation of Asn256, we noticed that in a determined AdoMet-bound SETD3 structure (PDB id: 3SMT), the side chain amide of Asn256 is hydrogen bonded to the main chain carbonyl group of Arg254. However, in our structure of AdoHcy-bound SETD3 with unmodified actin peptide, the side chain amide of Asn256 is hydrogen bonded to the main chain carbonyl group of Ala269 (Figure 2A). To validate the rigidity of the Asn256 side chain, we made two single mutants, N256D and N256Q. While N256Q bound to actin peptide 14-fold weaker than the wild type protein, N256D weakened the peptide binding affinity by ~130-fold. Thus, we agreed with the referees that the side chain of Asn256, which is rigidified by the hydrogen bond between Asn256 and Ala269, could not be flipped after catalysis. In our refined product-bound structure of SETD3, the carboxyl group of Asn256 faces toward the His73 imidazole ring, forming one hydrogen bond with the N^1^ of His73 (Figure 2B).

We have now followed the referees’ suggestions by removing the catalytic mechanism figure and minimizing the discussion about the catalytic mechanism.

2) The methyltransferase activity assay used here only measures relative rates. More detailed kinetic analysis is needed to discriminate binding effects from catalysis. If the substrate Km values are too high to determine, then this should be clearly indicated, and the limitations of the measurements stated more directly to readers. Furthermore, specific activities presented as V/E (velocity over enzyme concentration in min-1) should be shown.

We performed more detailed kinetic analysis for SETD3 using full length recombinant human β-actin as the substrate and determined the V_max_, K_M_, and k_cat_ (V/E) As shown in a new Table 3, all mutants display clearly lower catalytic efficiency (k_cat_/K_m_) towards actin and AdoMet than the wild-type enzyme. Among them, R75A and N278A are almost catalytically inactive variants of SETD3, confirming the importance of Arg75 and Ans278 in binding to AdoMet binding.

Both Asn256 and Tyr313 of SETD3 are catalytic pocket residues and play an important role in catalysis as well as substrate binding. Taking into account that these residues interact mainly with His73 of _β_-actin, and not with AdoMet (Figure 2A-2B and Figure 1—figure supplement 2A), we suggest that Asn256 and Tyr313 of SETD3 are plausibly required to keep the imidazole ring of His73 in the optimal position to facilitate the efficient transfer of the methyl group from AdoMet to His73. The consequence of the N256A and Y313F mutations is thus a decrease in the rate of “successful” methylation events that could be partially overcome by an increase in AdoMet concentration, as evidenced by the much lower k_cat_ values of both mutants (Table 3).

We have now introduced a short paragraph into the Results section to provide information about kinetic properties of the wild-type and mutant SETD3 proteins.

3) The writing of the paper and its presentation needs major improvements. Key figures are placed in the supporting information whereas figures that minimally add value are included with the main figures. For example, Figure 2 could either be in the supporting info or be one panel of a multi-paneled figure. Figure 4 is of limited impact. The comparisons to LSMT and CARNMT1 could also be one panel of a multi-paneled figure. By contrast, the table that reports the effects of mutations on peptide binding is in the supporting info--this should be in the main section. Likewise, the electron density maps that depict the methylated histidine are in the supporting figures and this should also be a main figure. The activity data are in the supporting info and this should be in the main section.The manuscript should be extensively edited for proper sentence structure, grammar and spelling to improve readability and comprehension. There are many convoluted sentences.

According to referees’ suggestions, we extensively improved the writing and presentation of the paper. We place key figures in main figures. Old Figure 2 is now placed as Figure 1—figure supplement 6. The comparisons to LSMT and CARNMT1 is now in Figure 2 (Figure 2D-2E) of revised manuscript. The table that reports the effects of mutations on peptide binding is now Table 1. We revise Figure 2A and Figure 2B to show the electron density maps of His73 and His73me, respectively, which discriminate His73me from unmodified His73. All kinetic parameters of SETD3 and its mutant are shown in Table 3 in main section. The manuscript was extensively edited for proper sentence structure, grammar and spelling to improve readability and comprehension, with convoluted sentences corrected. The formats of figures are revised according to *eLife*’s requirement.

4) Because SETD3 recognizes a long β-actin peptide (residues 66-88), the authors should really test the activity and binding affinity of the enzyme using comparable length H3K4 and H3K36 peptides. The H3K4 (1-10) and H3K36 (31-44) peptides tested may have been too short for binding and methylation by SETD3. If this will be too time consuming, at the very least the authors should state this caveat in their interpretation of the comparative enzymologic studies.

We performed ITC binding experiments with two longer histone peptides, H3K4(1-23) and H3K36(25-47) and found that neither of them displays detectable binding to SETD3 (Table 1). In addition, we tested by mass spec the activity of SETD3 towards two longer histone peptides and found that neither of them could be methylated by SETD3 (Figure 1—figure supplement 1).

We have now introduced a few sentences into the subsection “SETD3 binds to and methylates β-actin”that provide information on the SETD3 activity towards longer histone peptides.